# Quality of Life in Cervical Cancer Survivors Treated with Concurrent Chemoradiotherapy

**DOI:** 10.3390/medicina59040777

**Published:** 2023-04-17

**Authors:** Raminta Stuopelytė, Guoda Žukienė, Rūta Breivienė, Vilius Rudaitis, Daiva Bartkevičienė

**Affiliations:** 1Faculty of Medicine, Vilnius University, M.K. Ciurlionio Str. 21, LT-03101 Vilnius, Lithuania; 2Clinic of Obstetrics and Gynaecology, Faculty of Medicine, Institute of Clinical Medicine, Vilnius University, Santariskiu Str. 2, LT-03101 Vilnius, Lithuania

**Keywords:** cervical cancer, concurrent chemoradiotherapy, quality of life

## Abstract

*Background and Objectives*: Cervical cancer is the fourth most common cancer in women globally. As survival rates gradually increase, it becomes necessary to assess the quality of life (QoL) after treatment. It is known that different treatment modalities have different effects on QoL. Therefore, we aimed to evaluate the QoL of cervical cancer survivors (CCSs) treated with concurrent chemoradiotherapy (CCRT). *Materials and Methods*: A cross-sectional monocentric study, conducted in Vilnius university hospital Santaros klinikos between November 2018 and November 2022, included 20 women, who were interviewed once using the European Organization for Research and Treatment of Cancer (EORTC)-designed Quality-of-Life questionnaire cervical cancer module (QLQ-CX24). The sociodemographic and clinical data as well as the results of the questionnaire are presented in mean, standard deviation and percentages. The QoL scores were compared between different age and stage groups using the Mann–Whitney U test. *Results*: Twenty participants, aged from 27 to 55 years, with a mean age of 44 years (SD = 7.6) participated in the study. All the participants were CCSs with an International Federation of Gynecology and Obstetrics (FIGO) stage from IB to IIIB and all of them were treated with CCRT. The symptom experience was relatively low and revealed a good result (21.8, SD = 10.2). Mean scores on body image, sexual/vaginal functioning, menopausal symptoms and sexual worry scales indicated moderate functioning and a moderate level of some of the cervical cancer specific symptoms after CCRT. Sexual activity and sexual enjoyment of the CCSs were low (11.7 (SD = 16.3), 14.3 (SD = 17.8), respectively). *Conclusions*: Cervical cancer survivors report a relatively good quality of life regarding symptom experience; however, women following concurrent chemoradiotherapy tend not to be sexually active and rarely feel sexual enjoyment. In addition, this treatment modality negatively affects a woman’s body image and self-perception as a woman.

## 1. Introduction

Cervical cancer is the fourth most common oncological disease among women globally. In 2020, 604,000 new cases and 342,000 deaths due to this type of cancer were identified worldwide [1]. The Baltic countries, including Lithuania, are among the European Union countries with the highest morbidity and mortality rates from cervical cancer [2,3]. The age-standardized incidence in 2020 in Lithuania was 18.7 cases per 100,000 inhabitants, and the European Union average was 12.8. The five-year prevalence of cervical cancer in all age groups in 2020 in Lithuania was 84.73 per 100,000 inhabitants [4].

According to the latest data, the overall five-year survival for all cervical cancer patients is 66%. The prognosis is poorer when the disease is advanced—a five-year survival rate is 16%. If cervical cancer is diagnosed at an early stage, the chances of survival are significantly better—92% [5].

Over the past 30 years, the introduction of cervical cancer screening programs has more than halved the incidence in countries with high gross domestic product (GDP) [6]. Mortality from cervical cancer in European countries is decreasing every year (from −0.2% to −2.7% per year) [2]. The overall 5-year survival rate is gradually increasing; therefore, the quality of life (QoL) after treatment becomes an important aspect in the selection of individual, patient-oriented treatment [5,6,7]. Due to broadening concepts of health beyond traditional indicators of morbidity and mortality, the need to measure QoL has become an integral part of patient care [8].

Depending on the stage of cervical cancer and any comorbidities, patients might be treated with surgery, radiotherapy, chemotherapy or the combination of all of the above methods [6,9]. Early-stage cervical cancer can be treated with surgery or radiotherapy, but most late-stage cancers require a combination of several treatments. Recent literature shows that the optimal method of treatment of locally advanced cervical cancer is concurrent chemotherapy with radiotherapy (CCRT) [10,11].

Clinical outcomes of cervical cancer, such as survival and health-related quality of life, are among the most important criteria for the effectiveness and quality of health care [12]. Research on health-related QoL in cervical cancer is particularly important [7]. All the mentioned treatment methods can have long-term adverse effects. However, there are some insufficient data showing that radiation treatment leads to a worse QoL, including impaired sexual function, compared to other treatment methods (surgery, chemotherapy) [13,14,15].

Scientists recognize that the diagnosis and treatment of cervical cancer has a negative impact on health-related QoL, and also has a profound psychological impact on a woman’s identity, which is closely related to her body image, self-confidence, satisfaction in social and intimate relationships, and her overall self-perception as a mother and a wife [7,15]. Therefore, the assessment of the QoL of cervical cancer survivors (CCSs) should include not only physical and functional wellbeing, but also emotional and social wellbeing.

More than 20 years ago, it was suggested that by regularly measuring the level of quality of life, the obtained data could potentially improve decisions about the prescribed cancer treatment and, at the same time, improve the outcomes for patients [16]. Nevertheless, the literature shows that much attention is paid to the prevention and treatment of cervical cancer, and only a small number of researchers focus on evaluating and improving the QoL of cervical cancer patients [12]. Although research on the QoL after cervical cancer treatment is gradually increasing and is being conducted in various countries, such as China, India, Tanzania, the Netherlands, Brazil, Japan and others, it has not yet been conducted in Lithuania [17,18,19,20,21,22]. Thus, we conducted a monocentric cross-sectional study aiming to evaluate the quality of life of cervical cancer survivors treated with concurrent chemotherapy with radiotherapy.

## 2. Materials and Methods

A descriptive cross-sectional monocentric study was conducted in Vilnius university hospital Santaros klinikos to assess the impact of CCRT on the QoL of cervical cancer survivors. The study took place between November 2018 and November 2022. The protocol was approved by the Vilnius Regional Biomedical Research Ethics Committee (approval No. 158200-18/10-1065-565, 2 October 2018).

The study included 20 participants who met the following criteria: were older than 18 years old, were diagnosed with cervical cancer stage I-III according to the International Federation of Gynecology and Obstetrics (FIGO) staging system and had agreed to participate in the study. Exclusion criteria included patients who were younger than 18 years old, were diagnosed with progressive cervical cancer, active infection, vesicovaginal or rectovaginal fistula and did not agree to participate in the study. All participants signed an informed consent form to be enrolled in this study.

The study consisted of two parts. In the first part, sociodemographic and clinical data (age, cervical cancer FIGO stage, applied treatment) was collected from the patients’ medical records. The second part analyzed QoL using a standardized and validated questionnaire on general and individual areas of quality of life. The questionnaires were distributed by the main researcher and given to the participants during outpatient consultations, and the participants were asked to fill in the questionnaires themselves.

The evaluation method selected for this study was the European Organization for Research and Treatment of Cancer (EORTC)-designed questionnaire, QLQ-CX24, to assess disease-specific and treatment-specific aspects of QoL in patients with cervical cancer. This questionnaire has been extensively tested in multicultural and multidisciplinary settings and has been confirmed to be reliable and valid [23]. The questionnaire used in this research was standardized and translated into the Lithuanian language, the scales of this questionnaire corresponded to good internal consistency indicators (Cronbach’s alpha 0.744–0.829) [24]. The EORTC-CX24 questionnaire consists of 3 multi-item scales and 6 single-item scales, forming a total of 24 questions. These items are divided into those assessing functioning (body image, sexual/vaginal functioning, sexual activity, sexual enjoyment) and those assessing symptoms (symptom experience, lymphedema, peripheral neuropathy, menopausal symptoms, sexual worry). The questionnaire uses a four-point response scale (not at all, a little, quite a bit, very much).

For model development, the categorical raw scale scores were linearly transformed to a score from 0 to 100, to process according to the EORTC scoring manual. A high score for a symptom scale/item represents a high level of symptomatology/problems, a high score for a functional scale represents a high/healthy level of functioning [25].

There were no missing values in the completed questionnaires. The questionnaire survey was conducted once. The timing of evaluation after CCRT differs for each patient. Sociodemographic and clinical data were calculated using descriptive statistics. The QLQ-CX24 scores of the scales evaluating symptoms were divided into three groups: good, if the score was ≤33.33, moderate 33.34–66.66, or poor ≥66.67. The scores of the functioning scales were divided into three groups: good, if the score was ≥66.67, moderate 33.34–66.66, or poor ≤33.33. The results are presented in mean, standard deviation and percentages. The continuous variables were tested for normality using the Shapiro–Wilk test. The nonparametric Mann–Whitney U test was used to compare the median scores of QoL scales between the examined groups of patients. A 5% level of statistical significance was used for variables (*p* < 0.05). Statistical analysis was performed using the Microsoft Excel 2016 MSO (Version 1901, Santa Rosa, California, USA) and IBM SPSS Statistics 26 (New York, USA) programs.

## 3. Results

Between November 2018 and November 2022, a total of 20 participants, all of whom were native Lithuanian speakers, aged from 27 to 55 years, with a mean age of 44 years (SD = 7.6) participated in a monocentric cross-sectional study. All the participants were cervical cancer survivors with a FIGO stage from IB to IIIB (Table 1). All patients were treated with concurrent chemoradiotherapy.

The symptom experience scale revealed a relatively good result with a mean score of 21.8 (SD = 10.2). As many as 90% (*n* = 18) of participants had a mean symptom score of ≤33.33, which indicates a relatively low frequency of physical symptoms and a relatively good level of QoL related to the occurrence of these symptoms. Only 10% (*n* = 2) of women reported experiencing symptoms moderately (symptom experience score 33.34–66.66), and none experienced them very much. In order to determine whether the occurrence of individual symptoms significantly predicts the estimated score of the symptom experience, a linear regression model was applied, and a multivariable analysis (ANOVA) was performed. The overall regression model was statistically significant: R^2^ = 0.969, F(6,13) = 66.862, *p* < 0.001. Multivariable analysis revealed that the score of symptom experience was most significantly affected by frequent urination (β = 0.2, SE = 0.021, *p* <0.001), lower back pain (β = 0.13, SE = 0.036, *p* = 0.003), irritation or soreness in the vagina or vulva (β = 0.15, SE = 0.026, *p* < 0.001), cramps in the abdomen (β = 0.08, SE = 0.033, *p* = 0.034) and discharge from the vagina (β = 0.06, SE = 0.025, *p* = 0.034).

In this questionnaire, the experiencing of lymphedema and peripheral neuropathy was evaluated separately from the overall symptom experience score. Among the participants, lymphedema and peripheral neuropathy experiencing was low (mean scores 6.7 (SD = 13.7) and 13.3 (SD = 16.8), respectively). However, a significant correlation between lymphedema and peripheral neuropathy occurrence (R = 0.61, *p* = 0.04) was found.

Menopausal symptoms: hot flashes and/or sweats were also evaluated separately. Among all participants, the mean score for these symptoms was higher than that for the previously described symptoms, at 48.3 (SD = 22.9), indicating an average occurrence of menopausal symptoms. A mild occurrence of menopausal symptoms (≤33.33) was reported by 45% (*n* = 9) of the participants, and 55% (*n* = 11) answered that they experienced hot flashes or sweating very much (menopausal symptoms score ≥ 66.67).

The last symptom, assessed separately from the overall symptom experience score, was worry about the painfulness of sexual intercourse. More than half of the participants, 60% (*n* = 12) answered that they felt very much worried about the pain of intercourse (sexual worry score ≥ 66.67). A total of 40% (*n* = 8) of the participants said that they felt this worry a little (≤33.33). The mean of the sexual worry score for all women was 58.3 (SD = 32.2) overall, indicating a moderate frequency of sexual worry.

Regarding body image, the mean of the scale scores of all participants for this factor was 51.1 (SD = 16.7), revealing a moderate level of dissatisfaction and lack of confidence in the appearance of their body. One third of the participants (30%, *n* = 6) felt less attractive, less feminine or very much dissatisfied with their body (body image score ≥ 66.67). Only 20% (*n* = 4) of women experienced dissatisfaction with their body image a little (≤33.33).

During the last four weeks before filling out the questionnaire, only seven participants (35%) were sexually active. The overall mean of the sexual activity score of all participants was 11.7 (SD = 16.3), revealing very low sexual activity among women treated with CCRT. In order to assess sexual and vaginal functioning, only participants who had been sexually active in the last 4 weeks were asked to answer the questions related to sexual/vaginal functioning. After calculating the overall mean of the scores for these questions, sexual/vaginal functioning was found to be at an average level (34.5, SD = 21.2). It was found that not a single participant reported good (sexual/vaginal functioning score ≥ 66.67) sexual/vaginal functioning, 57.1% (*n* = 4) indicated moderate (33.34–66.66), and 42.9% (*n* = 3) reported poor functioning (≤33.33). An equally relevant question for assessing sexual function is the question about the experience of sexual enjoyment. Of the sexually active women, all said that they did not feel sexual enjoyment at all or felt it only a little (sexual enjoyment score ≤ 33.33). The overall mean of the sexual enjoyment score indicated poor sexual functioning (14.3, SD = 17.8) (Table 2).

Participants were divided into two groups, according to the FIGO stage: early-stage disease (FIGO I) (*n* = 4) and advanced-stage disease (FIGO II-III) (*n* = 16). The differences between all the multi-item and single-item scale scores between the two groups were statistically insignificant (*p* > 0.05). According to the median participants’ age, we divided them into two groups: younger than 44 years old (50%, *n* = 10), and ≥44 years old (50%, *n* = 10). No statistically significant differences were found between the two age groups (*p* > 0.05) when evaluating the QoL.

## 4. Discussion

Post-treatment quality of life is an important factor to consider before treating patients with cervical cancer [19,26]. Patients with early-stage cervical cancer may be treated with more than one treatment method, so it is important that women understand the significant changes in QoL after treatment with each method. Long-term survival in young patients with cervical cancer highlights the weight of late adverse events, especially given the increasing number of younger patients [17,22].

The number of women treated for cervical cancer is on the increase, but the QoL of CCSs is poorer not only when compared with the healthy age-matched general population, but also when compared with other gynecological cancer survivors [27].

Radiotherapy (RT) is different from other treatment modalities, because it damages not only cancer cells, but also healthy cells around the tumor. Therefore, the surrounding tissues of a similar structure, which make up the cervix, uterine body, vagina, bladder, and rectum are also affected. Bladder and bowel disorders are more common after radiation therapy and can also cause ovarian failure in premenopausal women [13,14,26,28]. Besides the negative impact of the cancer and its treatment on the physical body, having cervical cancer also affects the woman psychologically and is expected to have profound implications for a woman’s self-identity and her body image [7].

Compared to the general population of healthy women, patients who have survived for more than 4 years after cervical cancer treatment report having a statistically significantly worse body image, greater worry about sexual relations, worse sexual and vaginal function, and they more frequently experience lymphedema and peripheral neuropathy [29]. However, such a negative effect on self-confidence and sex life may not only be a result of radiation, but also chemotherapy, and these methods, according to current recommendations, are often used together in the treatment of early-stage cervical cancer (such treatment was also administered to the participants of our study). It is known that the negative impact of radiation treatment on the QoL of a woman is much greater than that of chemotherapy [30,31]. When compared, the cervical cancer specific QoL scores of patients who received radiation and those who received chemoradiation treatment revealed no significant differences [32].

The aim of our study was to evaluate the effect of CCRT on QoL. The results of our study do not differ much from the results of other QoL studies described in the literature. The study revealed that experiencing cervical cancer specific physical symptoms is not the primary problem after CCRT. Nevertheless, we managed to distinguish the symptoms that have the most negative impact on the QoL: frequent urination, irritation or soreness in the vagina or vulva and lower back pain. However, of greater concern are the early occurrence of menopausal symptoms, and the scores of items related to the sexual functioning and body image perception. We found that the sexual activity and sexual enjoyment of the CCSs were particularly low and the scores for sexual worry, as well as dissatisfaction with one’s body image, were considerably high.

Ferrandina et al.’s study revealed slightly better results concerning cervical cancer specific QoL, except for lymphedema [7]. In the study, performed by Dahiya et al., post-treatment sexual activity (83.0, SD = 19.5) and enjoyment (63.6, SD = 27.7) scores were found to be significantly better than those in our study, and there was a very low occurrence of menopausal symptoms (1.2, SD = 6.2) [8]. In a similar study, Stanca et al. evaluated QoL 48 weeks after oncological treatment and obtained quite similar results related to symptom experience and functioning, except for lymphedema and peripheral neuropathy experiencing, which were found to be extremely high. However, it is necessary to mention that these results were calculated for patients treated with various methods and not exclusively with CCRT [33]. Another study of 90 participants demonstrated significantly worse functioning scores but significantly better symptom experience scores after chemoradiation compared with our study [26]. A study performed by Mvunta et al. included more than 300 participants and revealed an overall better QoL except for peripheral neuropathy and sexual activity [19].

It is important to pay attention to the fact that the manifestation of lymphedema and peripheral neuropathy does not appear immediately but, more often, at least several months after treatment [34]. The results of a study evaluating the QoL of women who survived more than 4 years after radiotherapy for cervical cancer suggest that the incidence of lymphedema and peripheral neuropathy increases with increasing duration after treatment [29]. Wiltink et al. conducted a systematic review of the impact of different cervical cancer treatment methods on QoL that only confirms this assumption. In addition to lymphedema and peripheral neuropathy, four to five years after treatment, dyspareunia, menopausal symptoms, difficulty with bowel control, urinary incontinence, or difficulty urinating are more common than immediately after treatment [35]. In addition, sexual worry increases over time [36]. Some authors suggest that symptoms associated with radiation therapy tend to increase up to ten years after treatment [37]. For some women, psychological repercussions remain, regardless of treatment modality, especially anxiety and worry associated with the potential for cancer recurrence and the disruption caused in daily life [38].

In our study, as in the studies of a considerable number of other researchers, no statistically significant differences in QoL scores were found between different age groups and between different stages of cervical cancer [18,26]. Some studies show that older age can lead to a worse perception of body image, a higher incidence of lymphedema and peripheral neuropathy. A higher FIGO stage may have a greater influence on the occurrence of menopausal symptoms [19]. In addition, there is evidence suggesting that patients with early-stage cervical cancer treated with radiation therapy are statistically significantly more active in their sexual life and experience sexual satisfaction more often compared to patients with locally advanced cervical cancer [7].

In our study, we did not compare the QoL between different treatment modalities, but there are some studies which did, and the findings of these studies support the results of our study. A population-based study by Korfage et al. revealed that women treated with radiotherapy experienced more symptoms, worse body image and more sexual worry (*p* < 0.05) than women treated with surgery [37]. This is confirmed in a review conducted by Pfaendler et al.—women who received radiation therapy complained of more bladder, bowel, and sexual dysfunction many years after treatment compared to surgically treated patients [39]. Dutch scientists compared QoL in cervical cancer survivors after primary surgery with QoL after primary radiotherapy and reached a conclusion that women treated with primary radiotherapy reported more physical and sexual symptoms than those treated with surgery [20].

Radiotherapy has a greater adverse effect on sexual function than radical hysterectomy with pelvic lymphadenectomy, but more recent research suggests that new radiotherapy methods may not so negatively affect sexual function in CCSs [27]. One of the ways to reduce the frequency of adverse events is to reduce the doses of ionizing radiation as much as possible. Yu et al. conducted a study to determine whether different doses (45-Gy and 50.4-Gy) of external beam radiation therapy differed in effect on QoL. The results revealed that appropriate dose reduction of external beam radiation therapy can reduce harmful effects on surrounding healthy organs without compromising clinical efficacy [40]. Another way to minimize the negative impact on QoL is to use combined radiation therapy—external together with brachytherapy. Mvunta et al. found in their study that when being treated with a combination of both radiation treatment methods, women experienced statistically significantly fewer specific symptoms, such as lymphedema, and peripheral neuropathy; they also experienced lower sexual worry, were more sexually active and had a better individual body image [19].

Of course, as mentioned earlier, it is not only the treatment that negatively affects the QoL, but also the disease itself. Therefore, it can be difficult to distinguish whether these symptoms are caused by the disease or adverse events that occur after the prescribed treatment. Dahiya et al. studied women’s QoL before and after chemoradiotherapy for cervical cancer. The results revealed a statistically significant reduction in physical symptoms and sexual worry after treatment, which should lead to improved QoL, but all other aspects assessed by the QLQ CX-24 questionnaire tend to worsen after treatment, with the greatest reduction in confidence in one’s body image [8]. Other studies also present similar results that several months after treatment, symptoms were significantly reduced, but vaginal and sexual function worsened [18,26].

Comparing the results of several studies, we noticed some variation, which may be due to the different number of participants and differences between participants, such as demographic or socio-cultural circumstances and local customs [41]. In any case, the variation in results is understandable because these types of studies rely on patient-reported outcomes and are subjective.

### Study Limitations

The questionnaire study we performed was monocentric and included a relatively small number of participants. It allowed us to see the tendencies in the QoL of CCSs and highlighted key issues for consideration when creating an individual treatment plan. In this study, we did not evaluate the influence of different treatment modalities or different radiation doses on QoL. Additionally, broader research could be performed to evaluate and compare the QoL and psychological state of patients after different durations of time following treatment. This could provide deeper and fundamental understanding about cervical cancer patient care in the long term after patients have reached remission. In order to reach definitive conclusions, we would recommend expanding the study to more centers and including more patients that have been treated for cervical cancer with CCRT.

## 5. Conclusions

This study described the main challenges of cervical cancer survivors following concurrent chemotherapy with radiotherapy. Cervical cancer survivors report a relatively good quality of life regarding symptom experience; however, after chemoradiotherapy, women tend not to be sexually active and rarely feel sexual enjoyment. In addition, this treatment modality negatively affects not only physical, but also psychological aspects of their status, especially self-perception as a woman and self-confidence in body image. In order to draw definitive conclusions about the effect of chemoradiation treatment on the quality of life, we recommend broader research.

## Figures and Tables

**Table 1 medicina-59-00777-t001:** Age and FIGO stage analysis.

Characteristics	Values, *n* (Percentage)
Age (years)	
≤30	1 (5.0%)
31–40	5 (25.0%)
41–50	9 (45.0%)
51–60	5 (25.0%)
Average age (years)	44 (SD = 7.6)
FIGO stage	
IB	4 (20.0%)
IIB	7 (35.0%)
IIIA	2 (10.0%)
IIIB	7 (35.0%)

FIGO—International Federation of Gynecology and Obstetrics.

**Table 2 medicina-59-00777-t002:** The QoL of the respondents.

Variables	Items	*n*	Mean Score	SD	95% CI	Scoring ≤33.33 (%) ^α^	Scoring 33.34–66.66 (%) ^α^	Scoring ≥66.67 (%) ^α^
QLQ-CX24 Symptom scale								
Symptom Experience *	31–37, 39, 41–43	20	21.8	10.2	17.0–26.6	90	10	0
Body Image #	45–47	20	51.1	16.7	43.3–58.9	20	50	30
Sexual/Vaginal Functioning #	50–53	7	34.5	21.2	14.9–54.1	42.9	57.1	0
Lymphedema *	38	20	6.7	13.7	0.3–13.1	100	0	0
Peripheral Neuropathy *	40	20	13.3	16.8	5.5–21.2	100	0	0
Menopausal Symptoms *	44	20	48.3	22.9	37.6–59.0	45	0	55
Sexual Worry *	48	20	58.3	32.2	43.3–73.4	40	0	60
QLQ-CX24 Functioning scale								
Sexual Activity #	49	20	11.7	16.3	4.0–19.3	100	0	0
Sexual Enjoyment #	54	7	14.3	17.8	−2.2–30.8	100	0	0

*n*—number of participants. SD—standard deviation. CI—confidence interval. * Scores range from 0 to 100, with a higher score indicating a greater experiencing of symptoms. # Scores range from 0 to 100, with a higher score indicating a higher level of functioning. ^α^ The part of participants who had corresponding score (≤33.33, 33.34–66.66 or ≥66.67).

## Data Availability

The data presented in this study are available on request from the corresponding author. The data are not publicly available due to privacy.

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
