# Peer review of "Quality of Life in Cervical Cancer Survivors Treated with Concurrent Chemoradiotherapy"

_medicina, 2023, doi:10.3390/medicina59040777_

Round 1
Reviewer 1 Report
The study of the quality of life in patients diagnosed and treated with cervical cancer is a subject with significant practical implications. The introduction is very brief. I recommend completing this section. The current article deals with a small number of patients. Therefore the conclusions are limited. I recommend completing the Discussions section with more bibliographic titles. Please check that in line 171, you refer to Korfage. That name is not found in the references.
Author Response
We agree with all the comments reviewer made. We tried to make corrections, to complete introduction and discussion with more bibliographic titles, and also we made a correction regarding Korfage citation.
Reviewer 2 Report
Although this paper is very well written and is on an important topic and is from a patient population that is under represented within the medical literature. There are however, major concerns, primarily around the very small participant numbers and participant selection.
How many cervical cancers does the Vilnius University Hospital see every year? Who decided which patients would be invited to participate? Consecutive cases or selected? 20 cases in 2 years doesn’t seem many cases. Who was distributing the questionnaire? Were all the participants primary Lithuanian speakers? Do you have any information about the characteristics of the participants who declined to participate? The study exclusion criteria will have filtered out patients with significantly greater issues and most likely lower QOL.
‘was assumed that the missing items would have had the 83 average of the items present’ – this assumption is not appropriate. It may be that these items were not included because they would be given a very low score and the participants didn’t want to disappoint the study team.
Why was the QLQ-CX24 scoring system not followed rather than dividing the results by 1/3rds.
For only 20 patients analysis to 2 decimal places is not necessary.
How many participants only partly completed the questionnaire? What were their demographics. Did only 7 participants complete the sexual enjoyment and functioning questions?
Although dividing the population into early/late stage and young/older is a good idea the number of participants is just too low to be meaningful, particularly when looking at sexual functioning.
The discussion is very light on discussion of the published literature on this subject.
Given the results presented I don’t see how the conclusion in the abstract is that ‘Cervical cancer survivors report a good quality of life regarding symptom experience’ surely this is quite the contrary. If the research had analysed the data using the official QLQ-CX24 scoring system they would come to a different conclusions.
Round 2
Reviewer 2 Report
The authors have improved the paper significantly with the changes that have been made. The discussion in particular puts the results in context and helps to mitigate for the small study numbers.
One aspect that has not been discussed however is the fear of cancer recurrence, stage at diagnosis and time from treatment for example DOI: 10.1111/ecc.13560. Given that these patients are 4 years out from treatment this is a good population to explore this in.
There does need to be a formal limitations section
Author Response
We cited an additional study in the discussion section, that second reviewer proposed to look into. We could not look into it during our study, because we followed a formal questionnaire. Of course, it would be beneficial to look into it in our future studies.
We agreed with the requirement to add a formal limitations section. So we did it.